# Perfect Associative Learning with Spike-Timing-Dependent Plasticity

**Christian Albers**
Institute of Theoretical Physics
University of Bremen
28359 Bremen, Germany
calbers@neuro.uni-bremen.de

**Maren Westkott**
Institute of Theoretical Physics
University of Bremen
28359 Bremen, Germany
maren@neuro.uni-bremen.de

**Klaus Pawelzik**
Institute of Theoretical Physics
University of Bremen
28359 Bremen, Germany
pawelzik@neuro.uni-bremen.de

## Abstract

Recent extensions of the Perceptron as the Tempotron and the Chronotron suggest that this theoretical concept is highly relevant for understanding networks of spiking neurons in the brain. It is not known, however, how the computational power of the Perceptron might be accomplished by the plasticity mechanisms of real synapses. Here we prove that spike-timing-dependent plasticity having an anti-Hebbian form for excitatory synapses as well as a spike-timing-dependent plasticity of Hebbian shape for inhibitory synapses are sufficient for realizing the original Perceptron Learning Rule if these respective plasticity mechanisms act in concert with the hyperpolarisation of the post-synaptic neurons. We also show that with these simple yet biologically realistic dynamics Tempotrons and Chronotrons are learned. The proposed mechanism enables incremental associative learning from a continuous stream of patterns and might therefore underlie the acquisition of long term memories in cortex. Our results underline that learning processes in realistic networks of spiking neurons depend crucially on the interactions of synaptic plasticity mechanisms with the dynamics of participating neurons.

## 1 Introduction

Perceptrons are paradigmatic building blocks of neural networks [1]. The original Perceptron Learning Rule (PLR) is a supervised learning rule that employs a threshold to control weight changes, which also serves as a margin to enhance robustness [2, 3]. If the learning set is separable, the PLR algorithm is guaranteed to converge in a finite number of steps [1], which justifies the term 'perfect learning'.

Associative learning can be considered a special case of supervised learning where the activity of the output neuron is used as a teacher signal such that after learning missing activities are filled in. For this reason the PLR is very useful for building associative memories in recurrent networks where it can serve to learn arbitrary patterns in a 'quasi-unsupervised' way. Here it turned out to be far more efficient than the simple Hebb rule, leading to a superior memory capacity and non-symmetric weights [4]. Note also that over-learning from repetitions of training examples is not possible with the PLR because weight changes vanish as soon as the accumulated inputs are sufficient, a property

which in contrast to the naïve Hebb rule makes it suitable also for incremental learning of associative memories from sequential presentation of patterns.

On the other hand, it is not known if and how real synaptic mechanisms might realize the success-dependent self-regulation of the PLR in networks of spiking neurons in the brain. For example in the Tempotron [5], a generalization of the perceptron to spatio-temporal patterns, learning was conceived even somewhat less biological than the PLR, since here it not only depends on the potential classification success, but also on a process that is not local in time, namely the localization of the absolute maximum of the (virtual) postsynaptic membrane potential of the post-synaptic neuron. The simplified tempotron learning rule, while biologically more plausible, still relies on a reward signal which tells each neuron specifically that it should have spiked or not. Taken together, while highly desirable, the feature of self regulation in the PLR still poses a challenge for biologically realistic synaptic mechanisms.

The classical form of spike-timing-dependent plasticity (STDP) for excitatory synapses (here denoted CSTDP) states that the causal temporal order of first pre-synaptic activity and then postsynaptic activity leads to long-term potentiation of the synapse (LTP) while the reverse order leads to long-term depression (LTD)[6, 7, 8]. More recently, however, it became clear that STDP can exhibit different dependencies on the temporal order of spikes. In particular, it was found that the reversed temporal order (first post- then presynaptic spiking) could lead to LTP (and vice versa; RSTDP), depending on the location on the dendrite [9, 10]. For inhibitory synapses some experiments were performed which indicate that here STDP exists as well and has the form of CSTDP [11]. Note that CSTDP of inhibitory synapses in its effect on the postsynaptic neuron is equivalent to RSTDP of excitatory synapses. Additionally it has been shown that CSTDP does not always rely on spikes, but that strong subthreshold depolarization can replace the postsynaptic spike for LTD while keeping the usual timing dependence [12]. We therefore assume that there exists a second threshold for the induction of timing dependent LTD. For simplicity and without loss of generality, we restrict the study to RSTDP for synapses that in contradiction to Dale's law can change their sign.

It is very likely that plasticity rules and dynamical properties of neurons co-evolved to take advantage of each other. Combining them could reveal new and desirable effects. A modeling example for a beneficial effect of such an interplay was investigated in [13], where CSTDP interacted with spike-frequency adaptation of the postsynaptic neuron to perform a gradient descent on a square error. Several other studies investigate the effect of STDP on network function, however mostly with a focus on stability issues (e.g. [14, 15, 16]). In contrast, we here focus on the constructive role of STDP for associative learning. First we prove that RSTDP of excitatory synapses (or CSTDP on inhibitory synapses) when acting in concert with neuronal after-hyperpolarisation and depolarization-dependent LTD is sufficient for realizing the classical Perceptron learning rule, and then show that this plasticity dynamics realizes a learning rule suited for the Tempotron and the Chronotron [17].

## 2   Ingredients

### 2.1   Neuron model and network structure

We assume a feed-forward network of $N$ presynaptic neurons and one postsynaptic integrate-and-fire neuron with a membrane potential $U$ governed by

$$\tau_U \dot{U} = -U + I_{syn} + I_{ext}, \tag{1}$$

where $I_{syn}$ denotes the input from the presynaptic neurons, and $I_{ext}$ is an input which can be used to drive the postsynaptic neuron to spike at certain times. When the neuron reaches a threshold potential $U_{thr}$, it is reset to a reset potential $U_{reset} < 0$, from where it decays back to the resting potential $U_\infty = 0$ with time constant $\tau_U$. Spikes and other signals (depolarization) take finite times to travel down the axon ($\tau_a$) and the dendrite ($\tau_d$). Synaptic transmission takes the form of delta pulses, which reach the soma of the postsynaptic neuron after time $\tau_a + \tau_d$, and are modulated by the synaptic weight $w$. We denote the presynaptic spike train of neuron $i$ as $x_i$ with spike times $t_{pre}^i$:

$$x_i(t) = \sum_{t_{pre}^i} \delta(t - t_{pre}^i). \tag{2}$$

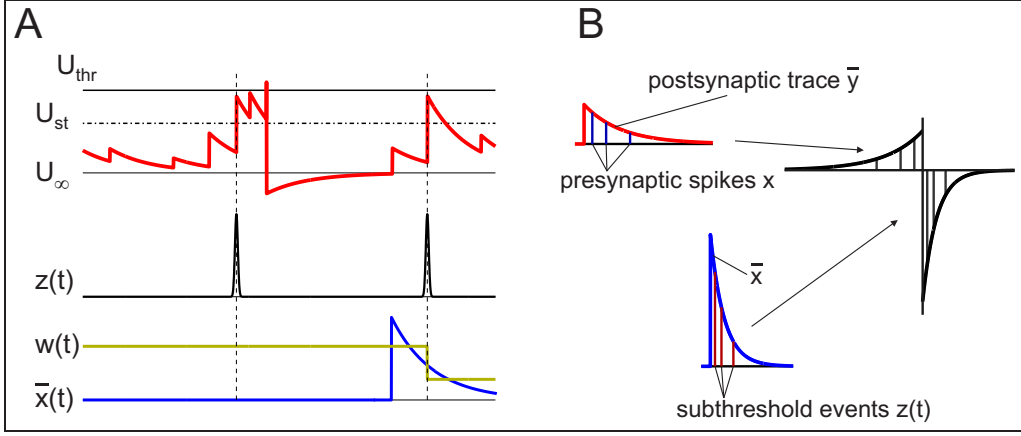

Figure 1: **Illustration of STDP mechanism**. **A**: Upper trace (red) is the membrane potential of the postsynaptic neuron. Shown are the firing threshold $U_{thr}$ and the threshold for LTD $U_{st}$. Middle trace (black) is the variable $z(t)$, the train of LTD threshold crossing events. Please note that the first spike in $z(t)$ occurs at a different time than the neuronal spike. Bottom traces show $w(t)$ (yellow) and $\bar{x}$ (blue) of a selected synapse. The second event in $z$ reads out the trace of the presynaptic spike $\bar{x}$, leading to LTD. **B**: Learning rule (4) is equivalent to RSTDP. A postsynaptic spike leads to an instantaneous jump in the trace $\bar{y}$ (top left, red line), which decays exponentially. Subsequent presynaptic spikes (dark blue bars and corresponding thin gray bars in the STDP window) "read" out the state of the trace for the respective $\Delta t = t_{pre} - t_{post}$. Similarly, $z(t)$ reads out the presynaptic trace $\bar{x}$ (lower left, blue line). Sampling for all possible times results in the STDP window (right).

A postsynaptic neuron receives the input $I_{syn}(t) = \sum_i w_i x_i(t - \tau_a - \tau_d)$. The postsynaptic spike train is similarly denoted by $y(t) = \sum_{t_{post}} \delta(t - t_{post})$.

## 2.2   The plasticity rule

The plasticity rule we employ mimics reverse STDP: A postsynaptic spike which arrives at the synapse shortly before a presynaptic spike leads to synaptic potentiation. For synaptic depression the relevant signal is not the spike, but the point in time where $U(t)$ crosses an additional threshold $U_{st}$ from below, with $U_\infty < U_{st} < U_{thr}$ ("subthreshold threshold"). These events are modelled as $\delta$-pulses in the function $z(t) = \sum_k \delta(t - t_k)$, where $t_k$ are the times of the aforementioned threshold crossing events (see Fig. 1 A for an illustration of the principle). The temporal characteristic of (reverse) STDP is preserved: If a presynaptic spike occurs shortly before the membrane potential crosses this threshold, the synapse depresses. Timing dependent LTD without postsynaptic spiking has been observed, although with classical timing requirements [12].

We formalize this by letting pre- and postsynaptic spikes each drive a synaptic trace:

$$
\begin{aligned}
\tau_{pre}\dot{\bar{x}} &= -\bar{x} + x(t - \tau_a) \\
\tau_{post}\dot{\bar{y}} &= -\bar{y} + y(t - \tau_d).
\end{aligned}
\tag{3}
$$

The learning rule is a read–out of the traces by spiking and threshold crossing events, respectively:

$$
\dot{w} \propto \bar{y}x(t - \tau_a) - \gamma\bar{x}z(t - \tau_d),
\tag{4}
$$

where $\gamma$ is a factor which scales depression and potentiation relative to each other. Fig. 1 B shows how this plasticity rule creates RSTDP.

## 3   Equivalence to Perceptron Learning Rule

The Perceptron Learning Rule (PLR) for positive binary inputs and outputs is given by

$$
\Delta w_i^\mu \propto x_0^{i,\mu}(2y_0^\mu - 1)\Theta\left[\kappa - (2y_0^\mu - 1)(h^\mu - \vartheta)\right],
\tag{5}
$$

where $x_0^{i,\mu} \in \{0,1\}$ denotes the activity of presynaptic neuron $i$ in pattern $\mu \in \{1,\ldots,P\}$, $y_0^\mu \in \{0,1\}$ signals the desired response to pattern $\mu$, $\kappa > 0$ is a margin which ensures a certain robustness against noise after convergence, $h^\mu = \sum_i w_i x_0^{i,\mu}$ is the input to a postsynaptic neuron, $\vartheta$ denotes the firing threshold, and $\Theta(x)$ denotes the Heaviside step function. If the $P$ patterns are linearly separable, the perceptron will converge to a correct solution of the weights in a finite number of steps. For random patterns this is generally the case for $P < 2N$. A finite margin $\kappa$ reduces the capacity.

Interestingly, for the case of temporally well separated synchronous spike patterns the combination of RSTDP-like synaptic plasticity dynamics with depolarization-dependent LTD and neuronal hyperpolarization leads to a plasticity rule which can be mapped to the Perceptron Learning Rule. To cut down unnecessary notation in the derivation, we drop the indices $i$ and $\mu$ except where necessary and consider only times $0 \leq t \leq \tau_a + 2\tau_d$.

We consider a single postsynaptic neuron with $N$ presynaptic neurons, with the condition $\tau_d < \tau_a$. During learning, presynaptic spike patterns consisting of synchronous spikes at time $t = 0$ are induced, concurrent with a possibly occuring postsynaptic spike which signals the class the presynaptic pattern belongs to. This is equivalent to the setting of a single layered perceptron with binary neurons. With $x_0$ and $y_0$ used as above we can write the pre- and postsynaptic activity as $x(t) = x_0\delta(t)$ and $y(t) = y_0\delta(t)$. The membrane potential of the postsynaptic neuron depends on $y_0$:

$$
\begin{aligned}
U(t) &= y_0 U_{reset} \exp(-t/\tau_U) \\
U(\tau_a + \tau_d) &= y_0 U_{reset} \exp(-(\tau_a + \tau_d)/\tau_U) = y_0 U_{ad}.
\end{aligned}
\tag{6}
$$

Similarly, the synaptic current is

$$
\begin{aligned}
I_{syn}(t) &= \sum_i w_i x_0^i \delta(t - \tau_a - \tau_d) \\
I_{syn}(\tau_a + \tau_d) &= \sum_i w_i x_0^i = I_{ad}.
\end{aligned}
\tag{7}
$$

The activity traces at the synapses are

$$
\begin{aligned}
\bar{x}(t) &= x_0 \Theta(t - \tau_a)\frac{\exp(-(t - \tau_a)/\tau_{pre})}{\tau_{pre}} \\
\bar{y}(t) &= y_0 \Theta(t - \tau_d)\frac{\exp(-(t - \tau_d)/\tau_{post})}{\tau_{post}}.
\end{aligned}
\tag{8}
$$

The variable of threshold crossing $z(t)$ depends on the history of the postsynaptic neurons, which again can be written with the aid of $y_0$ as:

$$
z(t) = \Theta(I_{ad} + y_0 U_{ad} - U_{st})\delta(t - \tau_a - \tau_d).
\tag{9}
$$

Here, $\Theta$ reflects the condition for induction of LTD. Only when the postsynaptic input at time $t = \tau_a + \tau_d$ is greater than the residual hyperpolarization ($U_{ad} < 0$!) plus the threshold $U_{st}$, a potential LTD event gets enregistered. These are the ingredients for the plasticity rule (4):

$$
\begin{aligned}
\Delta w &\propto \int \left[ \bar{y}x(t - \tau_a) - \gamma\bar{x}z(t - \tau_d) \right] dt \\
&= x_0 y_0 \frac{\exp(-(\tau_a + \tau_d)/\tau_{post})}{\tau_{post}} - \gamma x_0 \frac{\exp(-2\tau_d/\tau_{pre})}{\tau_{pre}}\Theta(I_{ad} + y_0 U_{ad} - U_{st}).
\end{aligned}
\tag{10}
$$

We shorten this expression by choosing $\gamma$ such that the factors of both terms are equal, which we can drop subsequently:

$$
\Delta w \propto x_0 \left( y_0 - \Theta(I_{ad} + y_0 U_{ad} - U_{st}) \right).
\tag{11}
$$

We expand the equation by adding and substracting $y_0\Theta(I_{ad} + y_0 U_{ad} - U_{st})$:

$$
\begin{aligned}
\Delta w &\propto x_0 \left[ y_0(1 - \Theta(I_{ad} + y_0 U_{ad} - U_{st})) - (1 - y_0)\Theta(I_{ad} + y_0 U_{ad} - U_{st}) \right] \\
&= x_0 \left[ y_0 \Theta(-I_{ad} - U_{ad} + U_{st}) - (1 - y_0)\Theta(I_{ad} - U_{st}) \right].
\end{aligned}
\tag{12}
$$

We used $1 - \Theta(x) = \Theta(-x)$ in the last transformation, and dropped $y_0$ from the argument of the Heaviside functions, as the two terms are seperated into the two cases $y_0 = 0$ and $y_0 = 1$. We do a

similar transformation to construct an expression $G$ that turns either into the argument of the left or right Heaviside function depending on $y_0$. That expression is

$$G = I_{ad} - U_{st} + y_0(-2I_{ad} - U_{ad} + 2U_{st}), \tag{13}$$

with which we replace the arguments:

$$\Delta w \propto x_0 y_0 \Theta(G) - x_0(1 - y_0)\Theta(G) = x_0(2y_0 - 1)\Theta(G). \tag{14}$$

The last task is to show that $G$ and the argument of the Heaviside function in equation (5) are equivalent. For this we choose $I_{ad} = h$, $U_{ad} = -2\kappa$ and $U_{st} = \vartheta - \kappa$ and keep in mind, that $\vartheta = U_{thr}$ is the firing threshold. If we put this into $G$ we get

$$\begin{aligned}
G =& I_{ad} - U_{st} + y_0(-2I_{ad} - U_{ad} + 2U_{st}) \\
=& h - \vartheta + \kappa + 2y_0 h + 2y_0 \kappa + 2y_0 \vartheta - 2y_0 \kappa \\
=& \kappa - (2y_0 - 1)(h - \vartheta),
\end{aligned} \tag{15}$$

which is the same as the argument of the Heaviside function in equation (5). Therefore, we have shown the equivalence of both learning rules.

## 4 Associative learning of spatio-temporal spike patterns

### 4.1 Tempotron learning with RSTDP

The condition of exact spike synchrony used for the above equivalence proof can be relaxed to include the association of spatio-temporal spike patterns with a desired postsynaptic activity. In the following we take the perspective of the postsynaptic neuron which during learning is externally activated (or not) to signal the respective class by spiking at time $t = 0$ (or not). During learning in each trial presynaptic spatio-temporal spike patterns are presented in the time span $0 < t < T$, and plasticity is ruled by (4). For these conditions the resulting synaptic weights realize a Tempotron with substantial memory capacity.

A Tempotron is an integrate-and-fire neuron with input weights adjusted to perform arbitrary classifications of (sparse) spike patterns [5, 18]. To implement a Tempotron, we make two changes to the model. First, we separate the time scales of membrane potential and hyperpolarization by introducing a variable $\nu$:

$$\tau_\nu \dot{\nu} = -\nu . \tag{16}$$

Immediately after a postsynaptic spike, $\nu$ is reset to $\nu_{spike} < 0$. The reason is that the length of hyperpolarization determines the time window where significant learning can take place. To improve comparability with the Tempotron as presented originally in [5], we set $T = 0.5s$ and $\tau_\nu = \tau_{post} = 0.2s$, so that the postsynaptic neuron can learn to spike almost anywhere over the time window, and we introduce postsynaptic potentials (PSP) with a finite rise time:

$$\tau_s \dot{I}_{syn} = -I_{syn} + \sum_i w_i x_i(t - \tau_a), \tag{17}$$

where $w_i$ denotes the synaptic weight of presynaptic neuron $i$. With $\tau_s = 3ms$ and $\tau_U = 15ms$ the PSPs match the ones used in the original Tempotron study. This second change has little impact on the capacity or otherwise. With these changes, the membrane potential is governed by

$$\tau_U \dot{U} = (\nu - U) + I_{syn}(t - \tau_d). \tag{18}$$

A postsynaptic spike resets $U$ to $\nu_{spike} = U_{reset} < 0$. $U_{reset}$ is the initial hyperpolarization which is induced after a spike, which relaxes back to zero with the time constant $\tau_\nu \gg \tau_U$. Presynaptic spikes add up linearly, and for simplicity we assume that both the axonal and the dendritic delay are negligibly small: $\tau_a = \tau_d = 0$.

It is a natural choice to set $\tau_U = \tau_{pre}$ and $\tau_\nu = \tau_{post}$. $\tau_U$ sets the time scale for the summation of EPSP contributing to spurious spikes, $\tau_\nu$ sets the time window where the desired spikes can lie. They therefore should coincide with LTD and LTP, respectively.

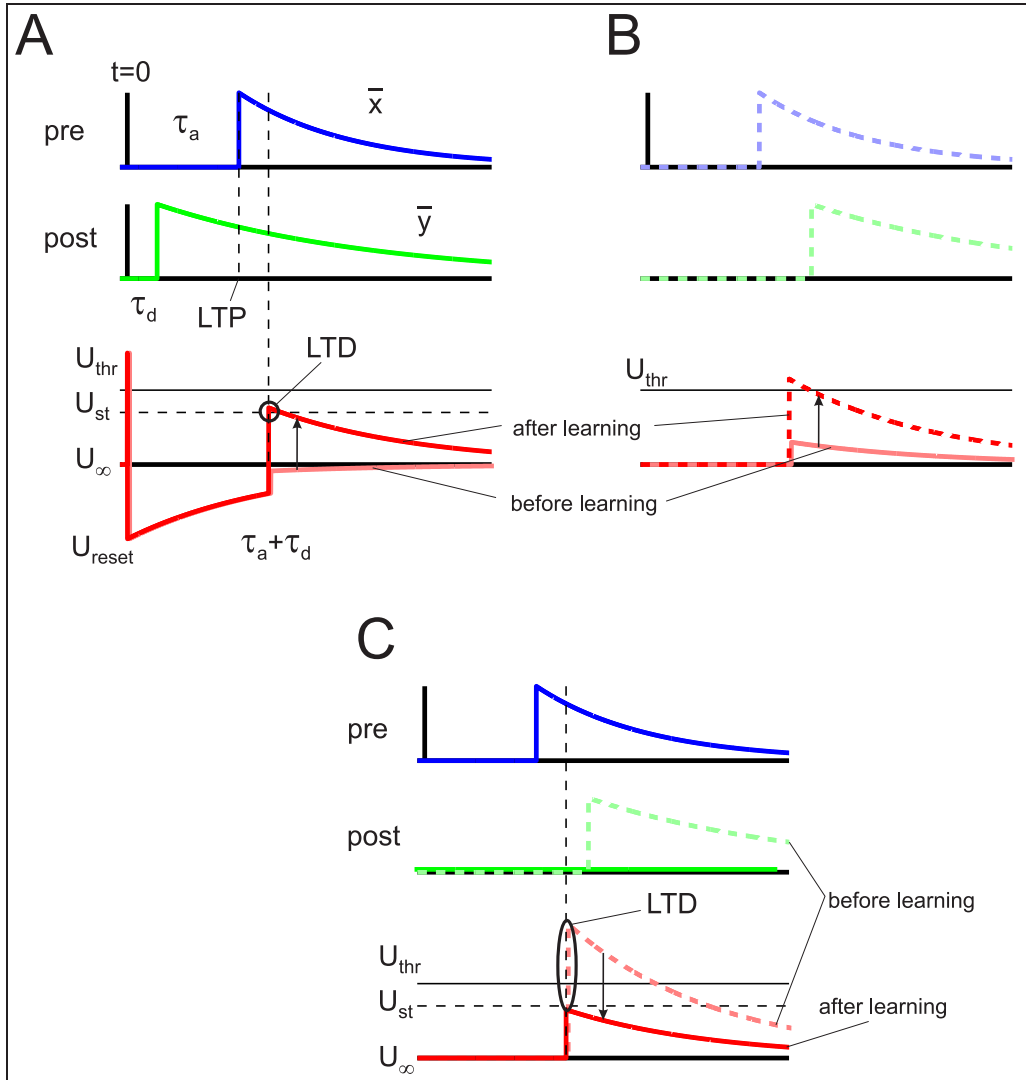

Figure 2: Illustration of Perceptron learning with RSTDP with subthreshold LTD and postsynaptic hyperpolarization. Shown are the traces $\bar{x}$, $\bar{y}$ and $U$. Pre- and postsynaptic spikes are displayed as black bars at $t = 0$. **A:** Learning in the case of $y_0 = 1$, i.e. a postsynaptic spike as the desired output. Initially the weights are too low and the synaptic current (summed PSPs) is smaller than $U_{st}$. Weight change is LTP only until during pattern presentation the membrane potential hits $U_{st}$. At this point LTP and LTD cancel exactly, and learning stops. **B:** Pattern completion for $y_0 = 1$. Shown are the same traces as in A at the absence of an inital postsynaptic spike. The membrane potential after learning is drawn as a dashed line to highlight the amplitude. Without the initial hyperpolarization, the synaptic current after learning is large enough to cross the spiking threshold, the postsynaptic neuron fires the desired spike. Learning until $U_{st}$ is reached ensures a minimum height of synaptic currents and therefore robustness against noise. **C:** Pattern presentation and completion for $y_0 = 0$. Initially, the synaptic current during pattern presentation causes a spike and consequently LTD. Learning stops when the membrane potential stays below $U_{st}$. Again, this ensures a certain robustness against noise, analogous to the margin in the PLR.

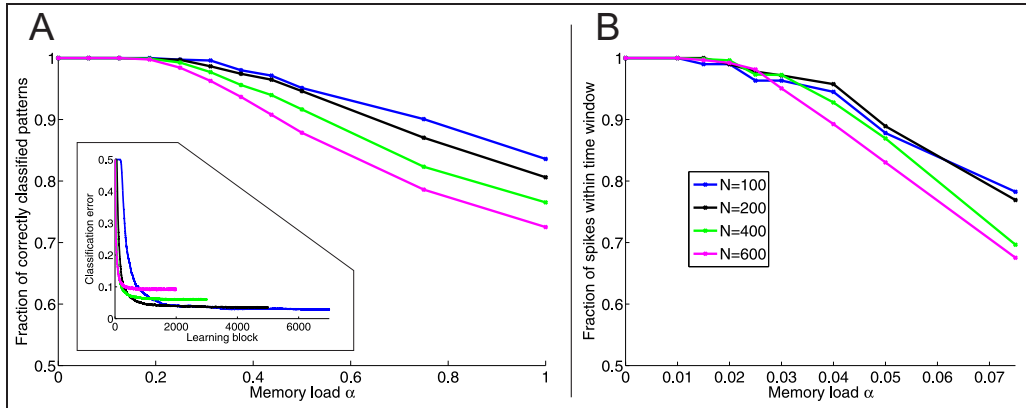

Figure 3: Performance of Tempotron and Chronotron after convergence. **A:** Classification performance of the Tempotron. Shown is the fraction of pattern which elicit the desired postsynaptic activity upon presentation. Perfect recall for all $N$ is achieved up to $\alpha = 0.18$. Beyond that mark, some of the patterns become incorrectly classified. The inset shows the learning curves for $\alpha = 7/16$. The final fraction of correctly classified pattern is the average fraction of the last 500 blocks of each run. **B:** Performance of the Chronotron. Shown is the fraction of pattern which during recall succeed in producing the correct postsynaptic spike time in a window of length 30 ms after the teacher spike. See supplemental material for a detailed description. Please note that the scale of the load axis is different in A and B.

Table 1: Parameters for Tempotron learning

| $\tau_U, \tau_{pre}$ | $\tau_\nu, \tau_{post}$ | $\tau_s$ | $U_{thr}$ | $U_{st}$ | $\nu_{spike}$ | $\eta$ | $\gamma$ |
|---|---|---|---|---|---|---|---|
| 15 ms | 200 ms | 3 ms | 20 mV | 19 mV | -20 mV | $10^{-5}$ | 2 |

### 4.1.1 Learning performance

We test the performance of networks of $N$ input neurons at classifying spatio-temporal spike patterns by generating $P = \alpha N$ patterns, which we repeatedly present to the network. In each pattern, each presynaptic neuron spikes exactly once at a fixed time in each presentation, with spike times uniformly distributed over the trial. Learning is organized in learning blocks. In each block all $P$ patterns are presented in randomized order. Synaptic weights are initialized as zero, and are updated after each pattern presentation. After each block, we test if the postsynaptic output matches the desired activity for each pattern. If during training a postsynaptic spike at $t = 0$ was induced, the output can lie anytime in the testing trial for a positive outcome. To test scaling of the capacity, we generate networks of 100 to 600 neurons and present the patterns until the classification error reaches a plateau. Examples of learning curves (Classification error over time) are shown in Fig. 3. For each combination of $\alpha$ and $N$, we run 40 simulations. The final classification error is the mean over the last 500 blocks, averaged over all runs. The parameters we use in the simulations are shown in Tab. 1. Fig. 3 shows the final classification performance as a function of the memory load $\alpha$, for all network sizes we use. Up to a load of $0.18$, the networks learns to perfectly classify each pattern. Higher loads leave a residual error which increases with load. The drop in performance is steeper for larger networks. In comparison, the simplified Tempotron learning rule proposed in [5] achieves perfect classification up to $\alpha \approx 1.5$, i.e. one order of magnitude higher.

## 4.2 Chronotron learning with RSTDP

In the Chronotron [17] input spike patterns become associated with desired spike trains. There are different learning rules which can achieve this mapping, including E–learning, I–learning, ReSuMe and PBSNLR [17, 19, 20]. The plasticity mechanism presented here has the tendency to generate postsynaptic spikes as close in time as possible to the teacher spike during recall. The presented learning principle is therefore a candidate for Chronotron learning. The average distance of these

spikes depends on the time constants of hyperpolarization and the learning window, especially $\tau_{post}$. The modifications of the model necessary to implement Chronotron learning are described in the supplement. The resulting capacity, i.e. the ability to generate the desired spike times within a short window in time, is shown in Fig. 3 B. Up to a load of $\alpha = 0.01$, the recall is perfect within the limits of the learning window $\tau_{lw} = 30ms$. Inspection of the spike times reveals that the average distance of output spikes to the respective teacher spike is much shorter than the learning window ($\approx 2ms$ for $\alpha = 0.01$, see supplemental Fig. 1).

## 5 Discussion

We present a new and biologically highly plausible approach to learning in neuronal networks. RSTDP with subthreshold LTD in concert with hyperpolarisation is shown to be mathematically equivalent to the Perceptron learning rule for activity patterns consisting of synchronous spikes, thereby inheriting the highly desirable properties of the PLR (convergence in finite time, stop condition if performance is sufficient and robustness against noise). This provides a biologically plausible mechanism to build associative memories with a capacity close to the theoretical maximum. Equivalence of STDP with the PRL was shown before in [21], but this equivalence only holds on average. We would like to stress that we here present a novel approach that ensures exact mathematical eqivalence to the PRL.

The mechanism proposed here is complementary to a previous approach [13] which uses CSTDP in combination with spike frequency adaptation to perform gradient descent learning on a squared error. However, that approach relies on an explicit teacher signal, and is not applicable to auto-associative memories in recurrent networks. Most importantly, the approach presented here inherits the important feature of selfregulation and fast convergence from the original Perceptron which is absent in [13].

For sparse spatio-temporal spike patterns extensive simulations show that the same mechanism is able to learn Tempotrons and Chronotrons with substantial memory capacity. In the case of the Tempotron, the capacity achieved with this mechanism is lower than with a comparably plausible learning rule. However, in the case of the Chronotron the capacity comes close to the one obtained with a commonly employed, supervised spike time learning rule. Moreover, these rules are biologically quite unrealistic. A prototypical example for such a supervised learning rule is the Temptron rule proposed by Gütig and Sompolinski [5]. Essentially, after a pattern presentation the complete time course of the membrane potential during the presentation is examined, and if classification was erroneous, the synaptic weights which contributed most to the absolute maximum of the potential are changed. In other words, the neurons would have to able to retrospectivly disentangle contributions to their membrane potential at a certain time in the past. As we showed here, RSTDP with subthreshold LTD together with postsynaptic hyperpolarization for the first time provides a realistic mechanism for Tempotron and Chronotron learning.

Spike after-hyperpolarization is often neglected in theoretical studies or assumed to only play a role in network stabilization by providing refractoriness. Depolarization dependent STDP receives little attention in modeling studies (but see [22]), possibly because there are only few studies which show that such a mechanism exists [12, 23]. The novelty of the learning mechanism presented here lies in the constructive roles both play in concert. After-hyperpolarization allows synaptic potentiation for presynaptic inputs immediately after the teacher spike without causing additional non-teacher spikes, which would be detrimental for learning. During recall, the absence of the hyperpolarization ensures the then desired threshold crossing of the membrane potential (see Fig. 2 B). Subthreshold LTD guarantees convergence of learning. It counteracts synaptic potentiation when the membrane potential becomes sufficiently high after the teacher spike. The combination of both provides the learning margin, which makes the resulting network robust against noise in the input. Taken together, our results show that the interplay of neuronal dynamics and synaptic plasticity rules can give rise to powerful learning dynamics.

**Acknowledgments**

This work was in part funded by the German ministry for Science and Education (BMBF), grant number 01GQ0964. We are grateful to the anonymus reviewers who pointed out an error in first version of the proof.

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
