[Supplementary Material]

# Supplementary material for: Perfect Associative Learning with Spike-Timing-Dependent Plasticity

**Christian Albers**
Institute of Theoretical Physics
University of Bremen
28359 Bremen, Germany
calbers@neuro.uni-bremen.de

**Maren Westkott**
Institute of Theoretical Physics
University of Bremen
28359 Bremen, Germany
maren@neuro.uni-bremen.de

**Klaus Pawelzik**
Institute of Theoretical Physics
University of Bremen
28359 Bremen, Germany
pawelzik@neuro.uni-bremen.de

## 1   Chronotron learning with RSTDP, subthreshold LTD and Hyperpolarization

### 1.1   Introduction

Here, we present model and scenario used to generate the capacity curves for the Chronotron. The model for the Tempotron learning (equations (16) to (18) in the main article) is slightly modified. The main reason is that during presentation, synaptic depression which precedes every postsynaptic teacher spike will be induced at every iteration. This would prevent convergence and destroys desired system states with perfect recall. This is not a concern in the Tempotron learning, because the teacher spike always occurs before any presynaptic activity. Therefore, the major change to the model for the Chronotron is to add synaptic scaling acting only on the negative weights.

### 1.2   Model description

Spike trains are sums of $\delta$-pulses:

$$x(t) = \sum_{t_{pre}} \delta(t - t_{pre}) \ , \ \ y(t) = \sum_{t_{post}} \delta(t - t_{post}) \ . \tag{1}$$

The synaptic current is

$$I_{syn}(t) = \sum_i w_i x_i(t) \ . \tag{2}$$

As in the Tempotron, we neglect axonal and dendritic delays. The membrane potential is governed by equation (1) of the main article, which means that we discarded the variable $\nu$ from the Tempotron model. The external current is used to deliver the teacher spikes and consists of a suprathreshold delta pulse at the desired times. The plasticity rule (equations (3) and (4)) remains in place. Pattern presentation and association protocol is similar to the Tempotron case. There are $N$ presynaptic and one postsynaptic neurons. We generate $P = \alpha N$ different random patterns. In each pattern $\mu \in P$, each presynaptic neuron spikes exactly once at a fixed time uniformly drawn from the interval $[0, T]$. Each presynaptic activity pattern is assigned one postsynaptic spike time $t_{teach}^\mu$, at which during the pattern presentation (associative learning) a teacher spike is induced by a suprathreshold external current. The teacher spike time is drawn from a slightly smaller interval (see below). Learning is

Table 1: Parameters for Chronotron learning

| $\tau_U, \tau_{pre}$ | $\tau_{post}$ | $U_{thr}$ | $U_{st}$ | $U_{reset}$ | $\eta$ | $\gamma$ | $\beta$ |
|---|---|---|---|---|---|---|---|
| 10 ms | 10 ms | 20 mV | 19.5 mV | -20 mV | $10^{-6}/N$ | 1 | 0.05 |

organized in learning blocks. During each block, each pattern is presented once, with the order of presentation randomized for every block. The weights are updated after each pattern presentation. Due to the considerations presented above, we introduce an additional weight decay term, which acts only on the (currently) inhibitory synapses. We denote the set of negative weights by $W^I(t)$. After each learning block, the negative weights are slightly reduced proportionally to their respective magnitude:

$$\Delta w_i = \begin{cases} -\beta w_i & \text{for } w_i \in W^I(t) \\ 0 & \text{else .} \end{cases} \tag{3}$$

This simple form of synaptic scaling has the disadvantage that the decay depends on the number of patterns. However, we found that the results are very insensitive to the parameter $\beta$, which justifies this choice.

After each learning block and after the synaptic scaling, we present each presynaptic pattern without the teacher input and with plasticity turned off. The pattern is counted as correctly completed if a postsynaptic spike occurs in the time window $[t^\mu_{teacher}, t^\mu_{teacher} + \tau_{lw}]$, where $\tau_{lw}$ is a parameter which controls the length of the learning window. Because the postsynaptic spike can occur over a finite time window, we reduced the time interval the teacher spike times are drawn from to $[0, T - \tau_{lw}]$ to make sure correct association can be achieved by every pattern.

We choose the length of the presentation interval $T = 200ms$ and $\tau_U = 10ms$ to match the respective parameters in the original Chronotron study [1]. The length of the learning window $\tau_{lw} = 30ms$ is associated to the time constants of the STDP window. From the perspective of the learning task the Tempotron is really just a special case of the Chronotron with a very long learning window ($t^\mu_{teach} \equiv 0$, $\tau_{lw} = T$). To allow plasticity over the whole window, we seperated the time scale of hyperpolarization from the membrane time scale, and set $\tau_{post} = \tau_\nu \approx T$. For the Chronotron, the postsynaptic spike has to occur as soon as possible after the time of the teacher spike, which requires a short time constant of LTP, $\tau_{post}$. Compared to the Tempotron, this parameter choice sacrifices capacity for precision. The learning window we use is relatively long compared to the millisecond (or even submillisecond) precision which is achieved with the alternative learning rules (E-Learning, [1], ReSuMe [2], PBSNLR [3]). However, in our case the mean time difference of the actual output spike to the teacher spike is much shorter than the learning window, between 2 ms and 14 ms. Higher loads $\alpha$ lead to larger time mismatches (See Fig. 1). It was shown by Florian [1] that ReSuMe and his own unoptimized I-Learning rule both reach a capacity of around 0.02, to which our own plasticity rule is very close. With this load, the average distance of desired to actual spike is small ($\approx 2ms$). We have to mention that the highly optimized E-learning rule has a much higher memory capacity (0.2), however at the expense of biological plausibility.

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

Figure 1: Examples of average differences in time of spike produced during recall and the teacher spike. **A** shows the time differences for a low load of $\alpha = 0.01$. Here, regardless of $N$ the difference converges to 2 ms. **B** shows the same for a load of $\alpha = 0.04$. Shown are only the time differences for successful recall. The average difference converges to a higher value around 10 ms. The gaps at the beginning are due to the fact that the initial weights are zero, and therefore there are no spikes during recall.