[Reviews · NeurIPS 2013]

Submitted by Assigned_Reviewer_1

The authors consider associative learning in networks of spiking neurons, and argue that a form of STDP with postsynaptic hyper-polarization is equivalent to the perceptron learning algorithm. The basic form of STDP proposed by the authors relies on traces (similarly to Morrison, Diesmann & Gerstner, “Phenomenological models of synaptic plasticity based on spike timing”, Biol Cybern, 2008, 98, 459-478, which should have been mentioned here), and allows for both potentiation and depression of the synapse. The authors then introduce the perceptron learning rule (PLR) for binary variables, in a form where the weighted sum of inputs is compared to a threshold in order to determine the update. As is well known, the PLR is a supervised learning algorithm requiring a target to be specified at the post-synaptic site. Therefore, when the authors move to establish the equivalence of the PLR to STDP they demand that the post-synaptic neuron is forced to spike for a target of 1, and prevented from spiking when the target is 0. Using a hyper-polarization mechanism, and taking into account a second post-synaptic event, they show that the supervised PLR can be translated into an unsupervised update rule, given in eq. (14). Finally, the authors introduce an additional depression term into the STPD update, in order to mitigate the effects of post-synaptic spikes following inputs which should not lead to a spike.
Overall the authors have presented an interesting approach to mapping between physiologically motivated models such as STDP and their computational interpretation within a supervised learning context. As such the paper is interesting, and, as far as I am aware, novel. Several comments/questions are in place.
(1) The authors refer to learning throughout the paper. However, they do not address learning but rather memorization, as there is no discussion of generalization beyond the training set. Note also that the PLR depends on initial conditions, as it terminates update once perfect classification is achieved. (2) The authors take care to introduce the various time scales (dendritic, axonal) but no discussion is devoted to their impact on performance. Moreover, it is not clear to me how robust are the result in section 4 to the choice of these parameters (and others of which there are many). (3) The algorithm is defined in a context where the target for each neuron is known, even though it is translated into an associative form using a newly suggested form of hyper-polarization. It is not clear to me how the effects of the network would influence performance. In this sense, it seems that reward based update rules, based on the so-called three-component Hebbian updates (which have been experimentally supported), would be more effective, as they directly relate to a global error signal, absent in the present formalism.
Summary: A spike based learning algorithm requiring post-synaptic hyper-polarization is presented which is claimed to be equivalent to the perceptron algorithm. This suggests an interesting functional interpretation to the suggested plasticity.

Submitted by Assigned_Reviewer_5

The paper presents a learning rule for a simple spiking neuron model
based on STDP, where the STDP for excitatory synapses is anti-Hebbian
and STDP for inhibitory synapses is Hebbian (i.e., it is reversed in
terms of the classical rule). The authors provide a proof that the
learning rule implements the Perceptron Learning Rule (PLR) in a
specific setup with synchronous input spikes and a kind of teacher
forcing. They also show that a modified version of the rule is able to
learn spatio-temporal spike patterns to some extend, relating it to
tempotron learning. The authors claim that this learning scheme is
biologically plausible.

General assessment:

Quality: The paper addresses an interesting topic. There seems to be a
small bug in the theory, but this does not cause a big problem. One
problem I see is that the manuscript does not contain a comparison
to the simplified tempotron learning rule which is in my opinion at
least as biologically plausible as the proposed learning
scheme. Another concern is that the simulations for spatio-temporal
patterns are done with a modified neuron and plasticity model with
some quite ad-hoc modifications. So it is unclear how the initial
theoretical investigation corresponds to that.

Clarity: The paper is reasonably well written, but clarity could be
improved substantially (see below).

Originality: The approach is quite probably novel, although there
exists some previous work that has not been cited.

Significance: The paper seems to be of some relevance. Since the
authors do not point to any specific system where this setup could be
implemented, I currently do not see any biological evidence for it
apart from the fact that anti-Hebbian STDP has been reported.

Specific points:

1) Clarity of the presentation
1.1) In line 90, it is stated that neurons are recurrently connected. Yet, I don't' see any place in the paper where this is the case. If I understand correctly, both the theory and the simulations consider simple feed-forward architectures. This is confusing as well as an overstatement and should be corrected.
1.2) It is unclear to me why the authors first introduce the PLR with a {-1,1} coding and then switch to the {0,1} coding. This is just confusing since additional variable-symbols are needed. Just start with the {0,1} coding. I believe that in many text-books it is defined in this way anyways and the mapping between the codings is close to trivial.
1.3) Figure 2 is misleading. The PSPs in Fig. 2 (red traces) are alpha-shaped with a finite rise time. This is misleading since according to the definition the PSPs should really be single exponentials. It is confusing because it does not match the definitions of ~h and ~x in lines 184-186.
1.4) Parameters alpha and beta are used in two different contexts in lines 189 and 310-311.

2) Formal proof of equivalence
There is one point which maybe I did not understand correctly.
I understand that the margin \epsilon for the case of a desired output spike (y_0=1) can be achieved by the after-spike depolarization. This means that during training, because of the teacher spike, the membrane potential has to be somewhat larger than during testing.
But I do not see how the analogous case works if the neuron should not spike (y_0=0). In this case, the depolarization needed to elicit a spike during training is exactly the same as during testing. Hence, the neuron cannot learn to bring the depolarization below (threshold-margin). Therefore, using "s" in the right term of the equation in line 204 is probably not correct. It is again misleading that in line 198 the authors suggest that beta can be chosen freely. I think this is not the case.
I actually see no way to correct for this. The only thing that can be done is to state that the margin for the PLR can only be learned for positive examples (i.e. y_0=1).

3) Comparison to tempotron learning
The authors compare their learning rule to tempotron learning in the text, but they do not exhibit an experimental comparison. First, the authors state that the tempotron learning rule is biologically unrealistic (line 55), but this applies only to the full rule, not to the simplified one. In order to see whether the current rule has any advantages over that one, an experimental comparison is needed (possibly also a comparison to the full tempotron rule to see how much one looses).

4) Learning of spatio-temporal patterns
4.1) Several modifications to the original setup are used here which are not well motivated. For example, it is unclear why the hyperpolarization variable nu is introduced in line 284. I suspect it is needed because one wants to avoid that the output neuron spikes more than twice. Also the modification of the learning rule eq. (18) is not clear. It looks like a hack and a assume that the parameters have to be tuned carefully if something changes in the input statistics. I did not understand the explanation in lines 300-303.
4.2) Related to the last point. I am not sure whether the initial setup works for spatio-temporal patterns (therefore probably the modifications). The reason is that the second spike of the output neuron acts as a teacher for the following spikes. So even if the desired output is y_0=0, if the neuron erroneously spiked at time t, for input spikes at t'>t, it looks like this was actually a positive pattern. What is the value for "T" in line 320? This value deems critical to me.

5) Relevance
The learning setup is that the trained neuron is forced to fire at the same time as the synchronous inputs or before a spatio-temporal input pattern. The former condition (for synchronous input) seems somewhat plausible in some contexts, the latter much less so. In general, if the teacher spike sometimes comes after the input spikes (which seems to be the more plausible situation in vivo) then the system breaks down.
Anti-Hebbian STDP has been reported for distal excitatory synapses in pyramidal cells, but on the other hand ref [10] of the manuscript also reported that depolarization would abolish that or make it Hebbian. This would quite probably destroy the properties needed here.

Minor points:

6) Literature:
The authors may want to compare their rule to ReSume (e.g. Ponulak and Kasinski, Neural Computation 2010) and discuss the relation to the investigations in (Legenstein et al., Neural Computation 2005).

7) It is a bit problematic to define the PLR over gradient descent on eq (7) since this error is not differentiable at E=0.
8) In line x_0^i >= 0 is used, later x_0^i \in {0,1}.
9) line 360: "biologically highly plausible" is an overstatement.
10) line 367: "substantial capacity". This is not clear. What is a substantial capacity in this case? Comparison with tempotron capacity etc.
Summary: The paper addresses an interesting topic. One
problem I see is that the manuscript does not contain a comparison
to the simplified tempotron learning rule. Some simulations are done with a neuron and plasticity model with some quite ad hoc modifications. The paper does not contain much information to assess the biological relevance of the proposed learning scheme.


Submitted by Assigned_Reviewer_7

This is an interesting and well written paper describing how the reverse spike-time dependent plasticity in combination with spike afterhypolarization can provide a biological mechanisms for both Tempotron and Perceptron learning. This approach has two advantages. First, it does not rely on an explicit teacher signal as in Ref. 12. Second, it does not require neurons to solve the credit assignment problem with respect to the contributions ot its membrane potentials at a certain time in the past (as in the original Tempotron paper). No weaknesses are noted.
Summary: A well written and interesting paper describing the importance of reverse spike time dependent plasticity rules for learning. These mechanisms in the brain (refs 9-11) for both inhibitory and excitatory neurons. This paper helps suggest a computational function for these empirical observations.
Author Feedback

Author rebuttal: We thank the reviewers for their feedback, especially reviewer #2 who provided a lot of detailed and helpful critique. Reviewer #2 pointed out an error in the proof of equivalence: for y_0 = 0 no margin is learned. We found a surprisingly simple solution to this problem: Introducing a subthreshold LTD into the learning rule establishes the margin also for y_0 = 0, as well as unifies the learning rules used in both sections of the manuscript.

We now adress the other questions in order of their appearence.
To reviewer #1:
(1) The learning rule we present inherits all the properties of the PLR, and terminates as soon as the margin is reached. The margin assures a certain generalization, which is independent of initial conditions.
(2) \tau_a and \tau_d do influence the margin, as they set \tilde x and \tilde y. As long as \tau_pre and \tau_post are much larger, the influence of \tau_a/d is neglible. Originally, we did not check robustness against variation of parameters in section 4. In later simulations, however, we now find that learning is quite robust.
(3) Learning in many neuronal systems certainly depends also on reward signals. Here we demonstrate the general possibility of pure associative learning with a biologically realistic mechanism and provide a strict proof of equivalence to the Perceptron Rule for a specific case. It would be very easy to also include some modulatory signal. However, this would obscure our central point. We decided to remove the mention of recurrentness in section 2, as we only employ feed-forward networks. This should remove confusion regarding network effects.

Impact Score: To our knowledge this is the first time the artificial PLR was realized with plausible neurobiological mechanisms (see rev. #2 and #3).

To reviewer #2:
General assessment, Quality:

Provided that the reviewer refers to the implementation using voltage convolution by the simplified tempotron learning rule, we admit that we did not treat this rule. In fact, even though the capacity is reduced by a factor of ~2, it is still higher with this rule than in our realization of tempotron learning. However, we do not agree on the biological plausibility. In the simplified tempotron rule, the teacher signal is unspecified and presumably convoluted, while in our rule this is provided by a simple spike with subsequent hyperpolarization, which is a well-known phenomenon.

Regarding the modifications in section 4: Having the subthreshold LTD improves convergence for the tempotron-like learning, and as mentioned above we now use it in the PLR proof as well. We decided to seperate hyperpolarization and membrane time constant to allow for substantial weight changes over the whole interval (T = 500 ms, which is not critical), while keeping a realistically shaped EPSP.

Significance: Our learning rule is a novel concept that depends on the interplay of a synaptic plasticity rule and somatic dynamics. Experimental studies in general treat those mechanisms seperately, and even if considered in conjunction in plasticity experiments, the membrane potential is simply clamped at a fixed value. This necessarily leads to a neglect of dynamic effects. Therefore, if successful, our idea should spark new research looking into this interplay. Also, it underlines the relevance of core biological mechanisms (STDP, hyperpolarization) for learning in real neuronal systems.

Specific points:
1), 1.1) to 1.4) are correct, and we will change them in the revised version.

2) See above.

3) The parameters in section 4 (EPSP shape, T) are very similar to those used by Gütig and Sompolinski, 2006. We did this to make the results comparable. Our capacity is close to 1/4 N, which is about one order of magnitude from the 3 N, the original tempotron capacity, and therefore also smaller than with the simplified rule. However, .3 N is still substantial (sufficiently different from zero), e.g. if compared to the capacity of the Hopfield network.

4) 4.1) For \nu and subthreshold LTD, see above. We will provide an improved explanation.

4.2) and 5) After submission we found that teacher spikes could hold any fixed position in the interval. This condition also lifts the requirement of a long hyperpolarization. With these changes, and the subthreshold LTD, in simulations the learning is quite robust against the choice of parameters. We suppress erroneous second spikes by a strong bias towards LTD in this setup.

We thank the reviewers for the literature suggestions. We will look into these studies and discuss them as necessary.